# Evaluation of the Heat Produced by the Hydrothermal Liquefaction of Wet Food Processing Residues and Model Compounds

**Morgane Briand [1,2,3], Geert Haarlemmer [1,*], Anne Roubaud [1] and Pascal Fongarland [2]**

1 CEA LITEN, Université Grenoble Alpes, 17 Avenue des Martyrs, 38000 Grenoble, France;
briand.morgane.44@gmail.com (M.B.); anne.roubaud@cea.fr (A.R.)
2 CNRS, CPE-Lyon, Catalysis, Polymerisation, Processes, Materials (CP2M), Université de Lyon,
Université Claude Bernard Lyon 1, UMR 5128, 2 Avenue Albert Einstein, 69616 Villeurbanne, France;
pascal.fongarland@univ-lyon1.fr
3 French Environment and Energy Management Agency, 20 Avenue du Grésillé, 49000 Angers, France
* Correspondence: geert.haarlemmer@cea.fr

**Abstract:** Hydrothermal liquefaction has proven itself as a promising pathway to the valorisation of low-value wet food residues. The chemistry is complex and many questions remain about the underlying mechanism of the transformation. Little is known about the heat of reaction, or even the thermal effects, of the hydrothermal liquefaction of real biomass and its constituents. This paper explores different methods to evaluate the heat released during the liquefaction of blackcurrant pomace and brewers' spent grains. Some model compounds have also been evaluated, such as lignin, cellulose and glutamic acid. Exothermic behaviour was observed for blackcurrant pomace and brewers' spent grains. Results obtained in a continuous reactor are similar to those obtained in a batch reactor. The heat release has been estimated between 1 MJ/kg and 3 MJ/kg for blackcurrant pomace and brewers' spent grains, respectively. Liquefaction of cellulose and glucose also exhibit exothermic behaviour, while the transformation of lignin and glutamic acid present a slightly endothermic behaviour.

**Keywords:** waste; biomass; hydrothermal liquefaction; heat release; continuous reactor

## 1. Introduction

Wet waste streams include a wide variety of products such as food processing residues and sewage sludge but also the organic fraction of municipal solid waste. They are often used, or abandoned, in low-value applications. Concerning the food processing industry, by-products are often dried and used as animal feedstock [1]. However, not all food residues can be used as animal feed, and one of the alternatives would be the energy valorisation of these residues [2]. Currently, anaerobic digestion is the only process used to upgrade wet residues into a gaseous fuel. The treated residues have a high water content in common, which typically varies from 50 to above 90%. For management, sanitary and transportation reasons, resources are sometimes dewatered or even dried before being handled. However, even if dewatering and drying are possible for many feedstocks, it comes with a significant cost. In the case of hydrothermal liquefaction, these steps are no longer necessary. In fact, hydrothermal liquefaction can be applied to a large variety of resources to produce a high heating value biocrude with a lower oxygen content than initial feedstock [3–7]. Biocrude can be further upgraded through several steps into biofuels to replace petroleum-derived fuels [8–10].

The hydrothermal conversion takes place in the water phase and produces a hydrophobic product, making separation of the final products easier. The conversion takes place at temperatures between 250 °C and 374 °C and at pressures above the saturation pressure of

water to ensure that water remains in the liquid phase, typically above 100 bar. This process is particularly adapted to wet biomass as drying is no longer required, taking advantage of the properties of water under subcritical conditions [11–13]. The reduced viscosity, the dielectric constant of water and the increased reactivity facilitate the conversion of biomass. Biomass is hydrolysed, solubilised and further converted through numerous reactions, among which are decarboxylation and dehydration [14].

Even though the basic principles of hydrothermal liquefaction are well known, there are still some scientific questions and technological issues delaying the development of liquefaction reactors. This is partly due to competition in the use of biomasses but also due to technological and economic difficulties [15,16].

A large majority of the hydrothermal liquefaction experiments presented in the literature are performed in batch reactors. However, the extrapolation of these results to a larger scale is hazardous in the absence of any validation in a continuous pilot. In continuous experiments, the reactor can be preheated at the desired temperature, increasing the heating rate of the biomass slurry [7]. Major differences in the results between batch and continuous reactors can be attributed to the heating rate. Brand et al. [17] report an increase in the conversion of wood and cellulose due to the increase in the heating rate. A heating rate of 20 K/min leads to a higher oil yield compared to a transformation with a heating rate of 2 K/min. Biocrude is favoured by higher temperatures and by higher heating rates. Zhang et al. [18] also observed a decrease in solid residue from 24.7% to 6.7% by increasing the heating rate from 5 K/min to 140 K/min. In addition to a better oil yield, continuous liquefaction is also more energy efficient than the conversion in batch reactors. Heat integration and aqueous phase recycling also reduce energy costs and wastewater handling [19,20].

A lot of work has been carried out on model compounds and their conversion. Due to their nature, each of the constituting compounds of biomasses are expected to react differently. Hemicellulose is an amorphous hetero-polymer. It was reported to be one of the first constituents to hydrolyse below 200 °C [13]. Cellulose is a succession of glucose monomers linked with $\beta_{1\rightarrow4}$ glycosidic bonds. This bond is hard to hydrolyse making cellulose more resistant than hemicellulose. Lignin has a polyaromatic structure and is mainly constituted of three monomeric compounds (p-coumaryl alcohol, coniferyl alcohol and sinapyl alcohol) [13,21]. It plays a major role in vegetal cell construction and is also hard to hydrolyse. The variability in the fibre composition and structure results in differences in their reactivity. In addition to this lingo-cellulosic content, food residues also contain simple sugars, proteins and lipids [22]. Proteins are polymers constituted from different amino acids. They first depolymerize with peptide bond cleavage, and the amino acids are then released. These rapidly react further through decarboxylation and deamination. Lipids are used to describe fatty material as grease or oil. Triglycerides are the most abundant form of lipids and release fatty acids through the hydrolysis releasing the glycerol backbone.

Different methods have been developed for the estimation of the heat released and, more specifically, the heat of reaction and of hydrothermal reactions. The published literature reports mainly the heat of reaction of hydrothermal carbonisation. None of the published experimental work was carried out to compare reaction enthalpies, especially on different food residues. A review conducted by Pecchi et al. [23] examined the estimation of enthalpy by the various methods.

An energy balance on reactants and products can be implemented to know whether energy has been produced or applied to the reaction medium. By comparing the energy of formation of reactants and products, the enthalpy of liquefaction reactions can be calculated with Hess' law (see Supplementary Materials). Several uncertainties are introduced when estimating the reaction enthalpy by the enthalpy balance based on product characteristics. A non-negligible number of organic species are transferred to the aqueous or gaseous phase. The calculation of their higher heating value remains sometimes challenging, but neglecting this part in the total energy balance can lead to an overestimation of the heat of reaction up to 100 or even 500% [24]. Goudriaan et al. [25] estimated the heat of reaction of

wood to be $-0.25 \pm 3$ MJ/kg based on feedstock and biocrude analysis. The error margin is high and the conclusion on the enthalpy of the reaction is uncertain, especially if the heat of reaction is close to zero [26].

The heat of reaction can also be calculated from differential scanning calorimetry measurement (DSC) or differential thermal analysis (DTA). Berge et al. [27] investigated the energetics of the hydrothermal carbonisation of food waste, paper, cellulose and municipal waste. Hydrothermal carbonisation takes place at slightly lower temperatures than liquefaction and favours the production of char [28]. Based on the mass balance and the higher heating value of the products, they calculated that all reactions are exothermic under 250 °C. Pecchi et al. [29] also investigated the heat of carbonisation of food waste. Grape marc was carbonised under three temperatures (180 °C, 220 °C, 250 °C). The reaction enthalpy was calculated from the characterisation of reactants and products. The heating value of the aqueous phase was estimated as the sum of the relative heating value of dissolved organics. The heat released during the carbonisation of grape marc was estimated between 3 and 5 MJ/kg. By taking into account water formation, often neglected on enthalpy calculation, they speculate that the formation of water is one of the most important contributors to exothermicity, and its formation shifts the results towards more exothermic reactions.

Cuvilas et al. [30] worked on hydrothermal carbonisation (HTC) of pretreated wood. Hydrothermal carbonisation of woody biomasses has been observed to be exothermic with a heat of reaction estimated between $-2.53$ MJ/kg and $-3.47$ MJ/kg with respective residence times of 150 min and 350 min under 180 °C. The acid pretreatment facilitates dehydration and energy densification of the resulting char. It also reveals that the reaction becomes more exothermic after acidic pretreatment.

Recently Lozano et al. [31] worked on a new correlation between the enthalpy of the formation of hydrothermal products during liquefaction and the initial content and properties of the biomass. This model was developed based on the experimental characterisation of 30 woody biomasses and optimized over experimental results. Parameters such as fixed carbon and ash content were introduced to increase accuracy. The model presents an error with experimental data below 6.5%. However, the model has not been tested yet on other types of biomass.

The estimation of the heat of reaction has also been performed with experimental work. Wood and cellulose were tested on differential scanning calorimetry (DSC) by Funke et al. [24]. It was observed that temperature dependency is minimal above a certain temperature. Under 240 °C, they measured $-1.07$ MJ/kg (on dry ash free basis, daf) against $-1.08$ MJ/kg (daf) at 260 °C. This highlights the role of kinetics on exothermic reactions. As long as they have reached a certain activation temperature and given a sufficient reaction time, calculation of the heat of reaction is reproducible. They observed that the heat of the reaction for wood ($-0.79$ MJ/kg (daf)) was lower than that of cellulose at 240 °C.

This method was also investigated by Pecchi et al. [32]. Cellulose, wood and digestate were carbonized with a heating rate of 5 °C/min to 250 °C. All the hydrothermal carbonisations of the biomass returned exothermic reactions with a reaction enthalpy of $-0.88$ MJ/kg for cellulose, $-0.64$ MJ/kg for wood and $-0.25$ MJ/kg for digestate.

The liquefaction reactions of cellulose and hemicellulose under subcritical conditions are generally found to be exothermic [24,32,33]. The conversion of biomass with high ash and lignin content does not show such heat release. The thermal degradation of hemicellulose occurs at lower temperatures than for cellulose. This suggests that crystalline cellulose is harder to hydrolyse. The exothermic activity of the cellulose and hemicellulose was confirmed by the work of Ibbett et al. [33] on the liquefaction of lignin, cellulose and hemicellulose directly extracted from wheat straw. Hemicellulose also exhibited heat release around 250 °C and cellulose around 280 °C. By deconvolution, they concluded that the hydrothermal conversion of lignin is slightly exothermic. Moreover, they found that exothermic peak temperature related to hemicellulose transformation is influenced by the water to biomass ratio. Higher dilutions allow for a more effective glycosidic bond scission

in hemicellulose and lower the temperature of the exothermic peak. This was not observed for the cellulose conversion, probably due to its higher crystallinity.

Cheng et al. [34] suggest that the hydrothermal depolymerisation reactions of lignin are endothermic and favoured at higher temperatures. However, they do not give any values.

Other methods have been developed on laboratory scale reactors to calculate the heat of reaction at a larger scale. Merzari et al. [35] developed a 2 L bench scale reactor where they compared the final temperature of the agave pulp and organic fraction of municipal waste with water after the transitory phase of heating. They measured the heat of reaction of −3.9 MJ/kg for agave pulp and −7.3 MJ/kg for the organic fraction of municipal waste. Goudriaan et al. [25] also addressed this question, injecting directly in preheated water. They also found exothermic reactions, with a resulting reaction enthalpy of −1.0 MJ/kg.

Online temperature measurements on a batch reactor enable direct monitoring and a better understanding of transformation on model compounds and real biomass. Rolland et al. [36] showed that the hydrothermal liquefaction of spirulina algae produced a heat of reaction of −1.7 MJ/kg$_{bm}$. In view of developing a continuous process, the estimation of the heat of reaction is important.

Thermodynamic data on the hydrothermal liquefaction conversion are scarce, however they are necessary for scaling up to an industrial level. The heat release during the liquefaction reactions has received very little attention. This work will focus on experimental work on both batch and continuous reactors.

This paper presents three different methods to evaluate the heat of reaction of the hydrothermal liquefaction of blackcurrant pomace and brewers' spent grains. Methods were developed on a bench scale autoclave. Continuous experiments were also carried out to estimate the heat of reaction. The results were compared to the results obtained in the batch reactor. These results will provide a deeper insight into the role of the composition on the heat of reaction, but will also provide the thermochemical data necessary to evaluate the process at a larger scale.

## 2. Material and Methods

### 2.1. Resources

The experiments were performed using blackcurrant pomace (BCP) and brewers' spent grains (BSG) supplied by local manufacturers ("Les Vergers de Boiron" and "La Brasserie du Dauphiné", respectively). Table 1 gives the elemental and biochemical compositions of the substrates. Water content, ash content and higher heating values are reported. Lipids, proteins and fibres content were obtained from external laboratories, and were on dry biomass. The fibre analysis was based on the NDF/ADF/ADL method following the norm. This method estimates the non-digestible part of food (and animal feed in particular). It is a convenient analysis method that classes lignin with the fibres. BCP is a residue from berry pressing, mainly constituted by seeds and peels: it is a wet and fibre-rich biomass also containing a non-negligible amount of proteins and lipids. BSG is the residue remaining after the extraction of malt from germinated barley.

Microcrystalline cellulose, alkali lignin, glucose and glutamic acid were purchased from Sigma-Aldrich and used as received. Dilution was performed with demineralized water. The higher heating values were measured with a bomb calorimeter. The humidity was measured by the mass difference after drying the samples to 105 °C overnight until a stable mass was reached.

**Table 1.** Characterisation of blackcurrant pomace (BCP) and brewers' spent grains (BSG).

|  | BCP | BSG | Methods |
|---|---|---|---|
| HHV (MJ/kg dry matter) | 22.2 | 20.5 | Calorimeter Bomb |
| HHV (MJ/kg as received) | 10.5 | 5.6 |  |
| Moisture (wt%) | 52.5 | 72.5 | ISO 18134-2:2017 |
| Ash content (wt% dry matter) | $3.7 \pm 0.3$ | $3.0 \pm 0.2$ | NF V 18-101 |
| % C | 49.2 | 43.8 | Dry basis |
| % H | 6.5 | 6.5 | Dry basis |
| % N | 2.9 | 2.7 | Dry basis |
| % S | 0.2 | 0.2 | Dry basis |
| % O | 37.5 | 43.8 | Dry basis, by difference |
| Proteins (g/100 g) | $18.4 \pm 0.8$ | $19.9 \pm 0.8$ | MI MONU08 Kjeldahl |
| Lipids (g/100 g) | $20.7 \pm 0.8$ | $6.2 \pm 0.5$ | NF ISO 6492 |
| Fibres on dry basis |  |  |  |
| Hemicellulose (g/100 g) | 15.1 | 30.1 | ST NF V18-122 |
| Cellulose (g/100 g) | 16.0 | 11.5 | ST NF V18-122 |
| Lignin (g/100 g) | 16.5 | 3.3 | ST NF V18-122 |
| Total Fibres (g/100 g) | 47.6 | 44.9 | ST NF V18-122 |

## 2.2. Batch Reactor

Hydrothermal liquefaction experiments were performed in a 0.6 L stainless steel (type 316) stirred batch reactor (Parr Instruments Company, Moline, IL, USA), Figure 1. In a typical batch experiment, the reactor is filled with 300 g of biomass slurry prepared from 30 g dried ground biomass with 270 g demineralised water. Overcharging the reactor leads to excessive pressure increases and may influence the yields from the liquefaction reactions [5]. The dry matter content is the same for the biomass slurry used in the continuous experiments. As mentioned previously, due to the difficult handling of concentrated slurry on continuous runs, a water to biomass ratio of 9:1 is chosen for all the runs. The autoclave is leak tested, purged and pressurized to 1.3 MPa with nitrogen gas to guarantee sufficient pressure for gas analysis after the reaction. The total pressure inside the reactor during the experiment is a function of the initial nitrogen pressure, the reaction temperature, the amount of water vapour and gas produced. The reactor is heated by an external heater with a nameplate electrical power 2 kW. A stirring speed of 600 rpm was set for all experiments.

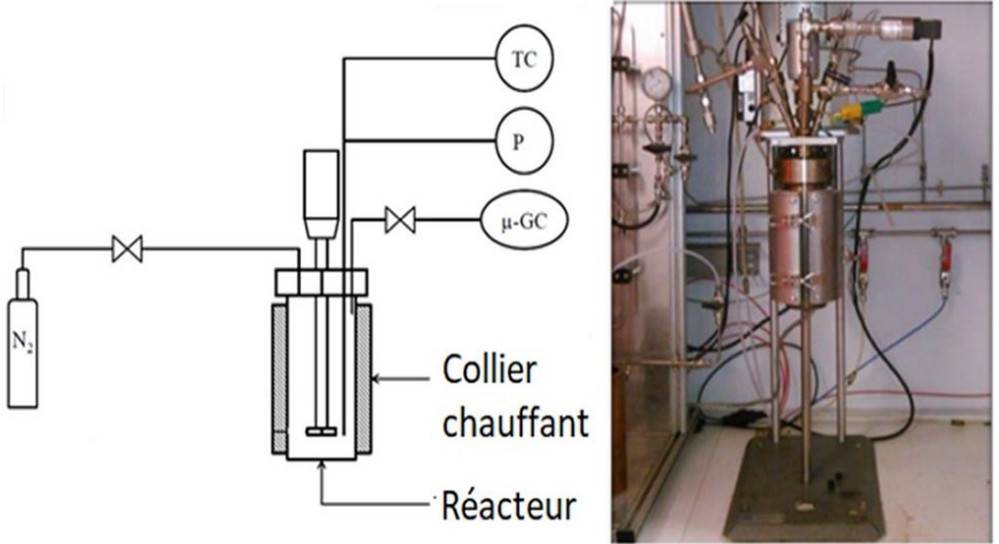

**Figure 1.** Batch Reactor (0.6 L, PARR).

The heater controller unit is very flexible, it allows different heating methods. The temperature is measured in the liquid. A temperature set point can be implemented and automatically reached by the controller. A temperature ramp can be applied to produce more reproducible results. The heater can also be controlled manually. After the holding time, the reactor was rapidly cooled down to room temperature in 30 min by an air quench. A thermocouple measures the temperature of the mixture.

### 2.3. Continuous Reactor

The continuous liquefaction pilot was a stirred 0.5 L stainless steel (type 316) tubular reactor. The reactor is fed by two alternating piston pumps. The principle of the equipment is shown in Figure 2. The feed preparation is stored in an agitated slurry tank (BM). The slurry is pumped (pumps P1) to high pressure and introduced in the reactor. The reactor is agitated with six propellers. The residence time for the experiment is about 20 min. Three external heating elements equipped with temperature controllers maintain a constant temperature in the reactor. As with the batch reactor, power can also be fixed manually. The pressure is regulated by two alternating piston pumps (P2). The products are collected in a closed tank. To avoid biomass sedimentation while feeding the reactor, xanthan gum is added as jellifying agent (2% on dry mass basis). The amount of xanthan gum is very low and has been shown in batch reactor experiments not to influence the reactions in any detectible way. The reactor operates at a flow rate of 1.5 L/h with a 10% biomass slurry. The approach was to fix the power in the heaters. The initial temperature settled at 290 °C for the inlet and 300 °C down the reactor. When the reactor was deemed sufficiently stable, the biomass slurry was injected. Gas is stored in the product tank, quantified and further analysed by a micro chromatograph. Biocrude is separated from the aqueous phase thanks to successive sieves of 200 and 50 μm.

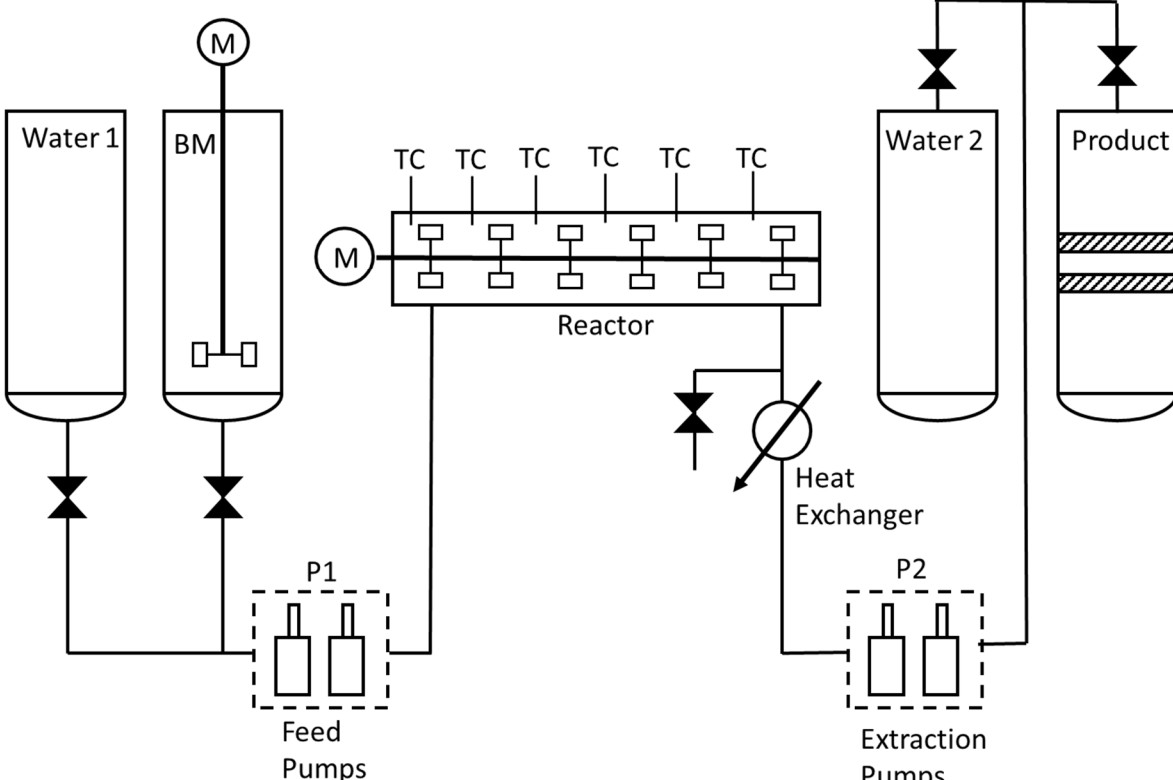

**Figure 2.** Continuous reactor (HYDROLIQ).

### 2.4. Products Recovery

After the experiments in the batch reactor, the pressure and temperature in the reactor was noted. The product recovery procedure is detailed in Figure 3. The gas was analysed by micro chromatography (μ-GC) to measure the volumetric composition. The reactor was opened after the gas analysis and biocrude was collected though filtration. Biocrude is the recovered product and is a mixture of an oily organic phase, the biooil and the biochar, which is a solid residue composed both of unconverted biomass and polymerisation compounds. In the case where biocrude looks like black powder, air circulation is sufficient for drying, but in the case of oily biocrude, further drying at 105 °C overnight is applied. The mass of the biocrude is corrected for the residual humidity, as measured by Karl Fischer titration.

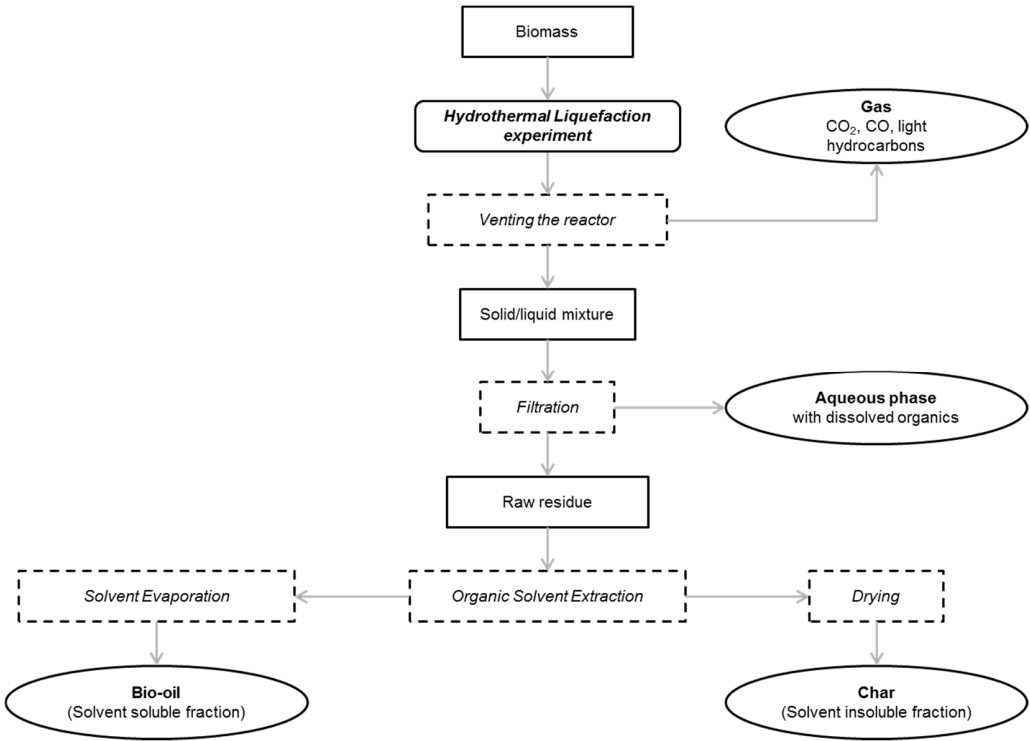

**Figure 3.** Recovery of products from batch hydrothermal liquefaction experiments.

The amount of gas formed is calculated with ideal gas law given its composition, pressure and temperature before and after each run. The ideal gas law underestimates the gas density by about 1% under these conditions compared to the Peng–Robinson or Soave–Redlich–Kwong equations of state.

Dissolved carbon dioxide is also taken into account with Henry's law. In a typical run, gas measurement showed a partial pressure $CO_2$ of 2–4 bar. Solubilisation and de-solubilisation of carbon dioxide is not supposed to be limited by kinetics. The aqueous phase is slightly acidic (pH 4 to 4.5) making it possible to approach the $CO_2$ dissolution ignoring the acid base reactions (and their pKa's) [37]. Under these conditions, molar concentration (C) of dissolved carbon dioxide can be calculated with Henry's law

$$C^l_{CO_2} = P_{CO_2} \cdot H_{CO_2, H_2O, T} \cdot \rho_{H_2O} \tag{1}$$

$$n_{dissolved_{CO_2}} = C^l_{CO_2} \cdot V_l \tag{2}$$

$C^l_{CO_2}$: molar concentration of dissolved $CO_2$ (mol·L$^{-1}$)
$P_{CO_2}$: Partial pressure of $CO_2$ (bar)
$\rho_{H_2O}$: Density of water (kg·L$^{-1}$)

$H_{CO_2,\ H_2O,\ T}$: Henry constant for a given temperature ($H_{CO_2,\ H_2O,\ 25°C}^0 = 0.034\ mol \cdot kg^{-1} \cdot bar^{-1}$) [38].

Henry's coefficient is highly dependent on temperature. The value retained is given for 298.15 K. In the case where gas was analysed under a different temperature, it can be adjusted according to the following equation

$$H_{CO_2,\ H_2O} = H_{CO_2,\ H_2O}^0 \cdot e^{\left(\frac{d\ \ln(H)}{d\frac{1}{T}} \cdot \left(\frac{1}{T} - \frac{1}{T^0}\right)\right)} \tag{3}$$

$\frac{d\ \ln(H)}{d\frac{1}{T}}$: 2400 (K) [39]

$T^0$: 298.15 (K)

The total amount of gas produced is calculated as the sum of gas measured from the pressure increase with ideal gas law and the amount of gas $CO_2$ dissolved in the water.

$$m_{gas_{produced}} = n_{gas_{measured}} \cdot \sum Mw_i x_i + n_{dissolved_{CO_2}} \cdot MCO_2 \tag{4}$$

$m_{gas_{produced}}$: Mass of gas produced (g)

$n_{gas_{measured}}$: Quantity of gas calculated by ideal gas law (mol)

$Mw_i$: Molecular weight of the species in the gas (g·mol$^{-1}$)

$x_i$: Molar fraction of the species *i*.

The final gas yield is defined as the ratio of the mass of gas produced over the mass of initial biomass introduced. Gas was also measured on the continuous pilot, and dissolved $CO_2$ was also taken into account in this case.

Once the reactor is opened, the aqueous phase is recovered after filtration on a Buchner funnel. An aliquot is diluted for the analysis of total organic and inorganic carbon with an analyser Shimadzu TOC-L CSH/CSN for the measurement of the total organic and inorganic carbon in the aqueous sample. To establish the complete energy balance, the energy contained in the aqueous phase has to be taken into account in the calculations. The aqueous phase is then dried in a rotary evaporator at 40 °C under vacuum (~300 mbar). The concentration of dissolved organics is calculated from the residue, as well as its elemental composition and its higher heating value.

The elemental composition of the dried biocrudes and organics dissolved in the aqueous phases was analysed with elemental analyser (Vario ElCube ELEMENTAR). Carbon, hydrogen, sulphur and nitrogen content were characterised. The oxygen content was deduced from the difference. In total, 10 mg of the sample was injected with tungsten oxide to catalyse the combustion at 1150 °C under an oxygen atmosphere. Combustion gas was vented to be reduced and analysed. The result gives the percentage of the mass of the element in the sample.

After a batch or continuous experiment, the gas retained in the reactor (batch) or product tank (continuous) is vented and analysed by a micro gas chromatograph (Varian Quad CP 4900, micro GC, Middelburg, the Netherlands) used online. Permanent gases ($O_2$, $H_2$, CO and $CH_4$) were analysed by a molecular sieve column using argon as a carrier gas. Light hydrocarbons ($C_2H_4$, $C_2H_6$, $C_2H_2$ and $C_3H_8$), $CO_2$ and sulphur species ($H_2S$ and COS) were analysed on a PoraPLOT U column using helium as a carrier gas. The quantity of gas formed during the experiment was calculated by initial and final temperatures and pressure measurements using the ideal gas law, and the composition of the gas phase obtained by micro chromatography.

### 2.5. Calculation of the Heat Release

Various strategies can be employed to estimate the heat released during the hydrothermal transformations. It is not possible to precisely evaluate the heat of each of the many reactions that take place, but it is possible to measure how much heat is released during the transformation. In a first approximation, we assume that the transformation is complete.

In all of the cases, the heat losses are supposed to be the same for the reference case and the biomass case, and differences were not taken into account. The differences in heating requirements due to variations in relative humidity of the nitrogen inerting the reactor were evaluated to be low (less than 0.05 MJ/kg of biomass) as the atmosphere has time to saturate due to the stirring during leak testing.

### 2.6. Batch Experiments with Temperature Ramp

The experiments were carried out on a suspension containing 10% dry feedstock. Results are compared with those obtained with an equal amount of water. The differences between the heat capacity of water and biomass (and its reaction products) can be compensated for in the calculations. We cannot be certain of the advancement of the reactions but the conversion was assumed to be complete. Different phases can sometimes be observed in the reactions (acceleration/deceleration of the temperature change), but, for the purpose of this paper, the overall heat effect is studied.

The temperature increased with a ramp of 15 K/min until the temperature set point. These experiments are very reproducible in terms of power control. The experiments were repeated four times.

The evolution of the temperature is identical in both cases. The difference in area between the two power curves is integrated and allows us to calculate the energy supplied by the reaction.

The energy liberated is calculated by

$$E_{Reaction} = \int \left( Power_{BM} - Power_{Ref} \right) \cdot dt \qquad (5)$$

$E_{Reaction}$: Energy of the reaction (J)
$Power_{BM}$: Power applied to the biomass case (W)
$Power_{Ref}$: Power applied to the reference case (W).

### 2.7. Batch Experiments with Imposed Power

A second method is to apply a rigorously identical power profile and observe the temperature evolution of the reaction mixture. In cases where a reaction is exothermic, final temperature $T_f$ of the reactor content should be higher than the reference final temperature $T_f^0$, under the same power supply, as biomass is supposed to provide energy to the reactional medium. On the contrary, in a case of endothermic reaction, the final temperature is expected to be inferior to the reference final temperature.

The energy supplied by the reaction corresponds to the energy required to heat up the reactor content and the reactor by $T_f - T_f^0$. The mass of the reactor is known, as well as the mass of biomass and water. The contribution to the overall heat capacity from the biomass is relatively modest compared to the water and steel. The heat capacity of biomasses and liquefaction products are unknown but have been taken to be equal to heat capacity of food residues (1.5 kJ·K$^{-1}$·kg$^{-1}$) [40] and are assumed to be constant under the different temperature considered.

With increasing temperatures, heat is absorbed by the heat capacity of water but also by its evaporation. In the case that the final temperature of the biomass case and the reference case are different, the final pressures are different and the amount of evaporated water will also be different. To estimate the difference, the energy required to heat water up to the measured temperature ($\Delta H_{water}$) was calculated in a process simulation program (ProSimPlus, [41]) for each case and each temperature. The simulation model is based on an isochoric calculation to calculate the amount of energy to heat water/nitrogen mixture at constant density (constant mass and volume in the autoclave reactor).

During these experiments, the full power (2 KW) was applied to the reactor furnace during a certain amount of time. Lower power was then set (400 W). This procedure does not reach a precise temperature, but it can compare the temperature obtained with water

with that obtained with biomass. Each of the experiments were performed two (three for pure compounds) times for each resource.

The energy liberated is calculated by

$$E_{Reaction} = \left(T_{Ref} - T_{BM}\right) \cdot (Cp_{Steel} \cdot m_{reactor} + C_{BM} \cdot m_{DB}) + \Delta H_{water} \tag{6}$$

$T_{Ref}$: Final temperature reached by the reference (K)
$T_{BM}$: Final temperature reached by the biomass (K)
$Cp_{Steel}$: Heat capacity of the reactor (kJ·K$^{-1}$·kg$^{-1}$)
$m_{reactor}$: Mass of the reactor (kg)
$m_{DB}$: Mass of the dry biomass (kg)
$C_{BM}$: Heat capacity of the dry biomass (kJ·K$^{-1}$·kg$^{-1}$)
$\Delta H_{water}$: Energy required to heat up water (J)

*2.8. Continuous Experiments with Imposed Power*

The same principle is used for continuous experiments. However, the biomass solution is injected directly in a preheated reactor and sufficient pressure is maintained to avoid partial vaporisation of water. The amount of energy required for temperature variations of the reactor wall is considered negligible in the calculation compared to the energy necessary to heat up the water.

The energy *E* liberated is calculated by

$$= \left(T_{Ref,} - T_{BM,}\right) \cdot (Cp_{Water} \cdot m_{Water} + Cp_{BM} \cdot m_{DB}) \tag{7}$$

In fixed power experiments the heating elements always operate at similar conditions. Even though there are small variations on the inside, we assume that the temperature on the outside of the heating element is relatively constant. The heat capacity of biomasses and liquefaction products are applied as described in the previous section. Due to the complexity of the experiments, only one run per resource was performed.

## 3. Results

*3.1. Batch Experiments with Controlled Temperature Ramp*

All the experiments with an imposed temperature ramp were carried out with a constant ramp of 15 K/min. Temperature and heater responses were recorded online. Figure 4 represents the heating curve for blackcurrant pomace.

The heat of reaction can be evaluated by integrating the differences in the applied power between the evaluated sample and a reference. The difference between the two power curves is caused by the heat liberated or absorbed during reactions. When we compare the temperature response of each of these batch experiments and that of water, we find that there are some major differences for the biomass solution with water. The actions of the temperature controller are those of a classic PID controller. In practice, with nearly identical environmental conditions, the evolution is very reproducible for a particular reaction mixture.

We observed a difference in temperature controller behaviour between the heating of the biomass solution and water. When a temperature ramp was imposed, the controller ramped up the output until it approached the final set point. After the setpoint was reached at the end of the ramp, the behaviour became rather unpredictable and the curves were no longer comparable. While this method produces a very repetitive heating curve, independent of the reactivity of the biomass, the fact is that the controller reaches 100% output at 260 °C for water, a little later for biomass. This means that we only observed the start of the reactions. Curves presented in Figure 4 are averaged on 12 experiments.

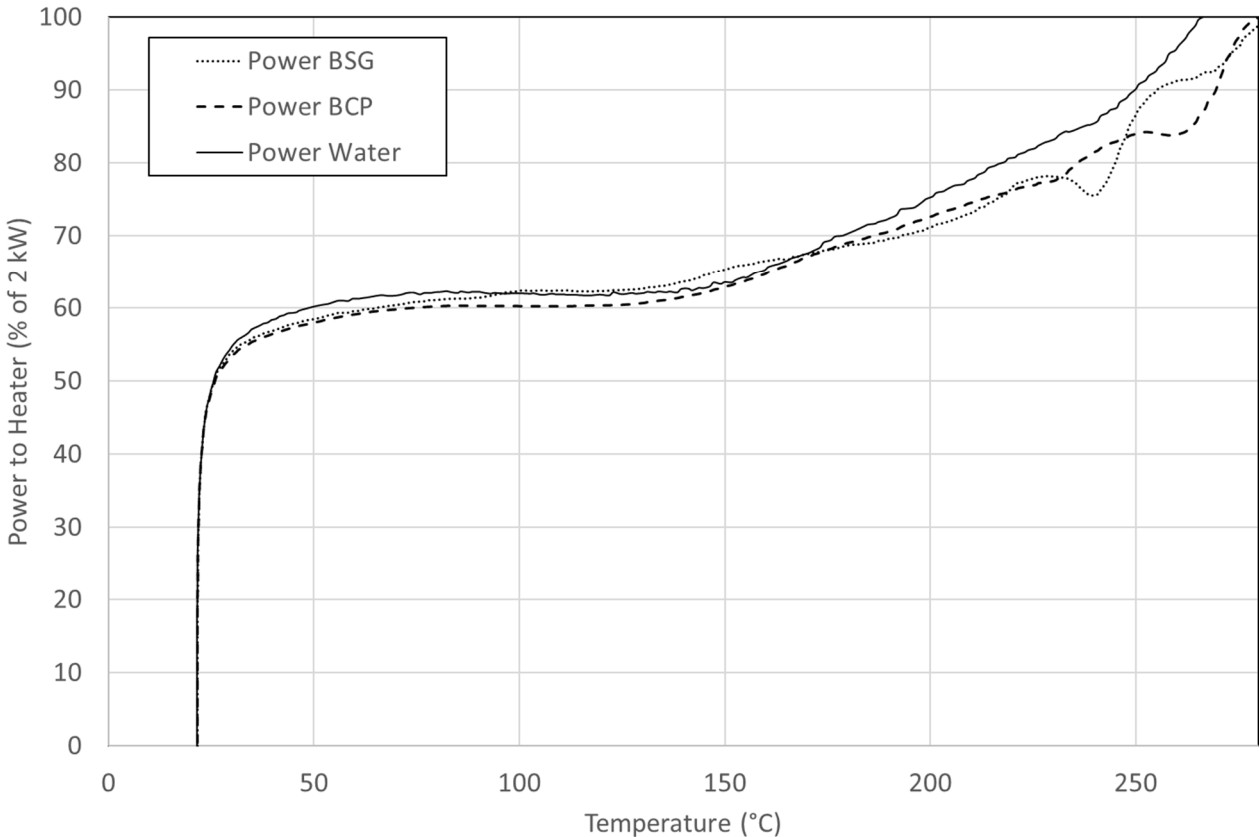

**Figure 4.** Average curve on power response to temperature ramp controlled for batch experiments on blackcurrant pomace (BCP), brewers' spent grains (BSG) and water as the reference.

The heat produced by the reaction of blackcurrant pomace is estimated at $\Delta H_{reac} = -1.4 \pm 0.3$ MJ/kg. This value is based on the heating up to 270 °C, as above this temperature no discrimination is possible. The heat of reaction is then underestimated. However, the online measurements of the temperature give some insights about the temperature when the heat is released. BSG has a similar behaviour as BCP during heating. The heat released has been estimated at $\Delta H_{reac} = -0.9 \pm 0.2$ MJ/kg$_{biomass}$.

For both resources, the reaction is mostly exothermic at 200 °C and higher. Peaks are observed at two moments. When the temperature first reaches 230 °C and again around 260 °C in the case of black currant pomace. For BSG, the first peak is more distinct whereas, for BCP, exothermicity is more visible at 250 °C. Noticeably, the composition does not differ much in cellulose content, but BCP shows a slightly higher exothermicity with this method.

The main difference between these two compounds may come from the difference in the composition, or in the reactivity of the initial compounds that may be affected by their crystallinity. However, the values remain close.

Elemental analysis of the biocrudes of BCP and BSG are presented in the form of the van Krevelen diagram in Figure 5 for 0-min holding time (i.e., the reaction is stopped at the moment the temperature is reached). In this case, the reaction was quenched the moment the reaction temperature was reached.

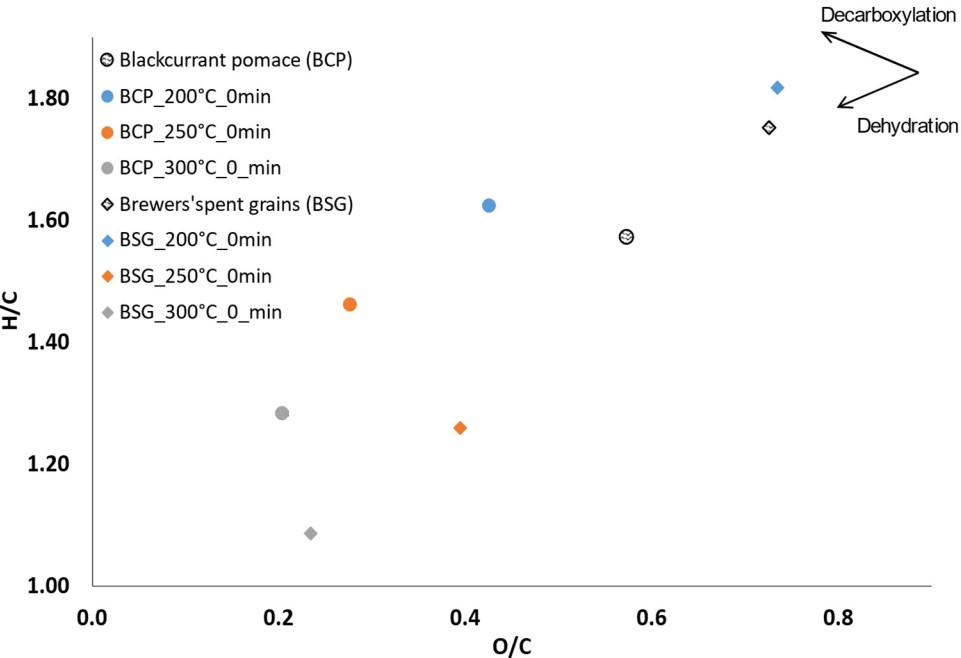

**Figure 5.** Van Krevelen Diagram for BCP (○) and BSG (◊) (0 min).

With increasing temperature, the concentration in oxygen and hydrogen decreases. During liquefaction, biomass releases oxygen through dehydration and decarboxylation reactions. With dehydration, deoxygenation occurs through the releasing of a molecule of $H_2O$. With dehydration reactions, the H/C ratio decreases sharply as well as the O/C ratio. In decarboxylation reactions, the H/C ratio rises slightly when the O/C ratio decreases [42]. According to the power curves (Figure 4), exothermic peaks occur around 230 °C and 260 °C for the two biomasses. Between 200 and 250 °C, the two biomasses undergo major transformation through dehydration. If dehydration plays a significant role in exothermic behaviour, as suggested by Pecchi et al. [29], this method does not allow full recording of the exothermicity, as the controller output saturates at 100%. The reaction is expected to be more exothermic than observed for the two residues. The final products seem to be quite similar for BCP and BSG in term of O/C and H/C ratios, although dehydration seems a little higher for BSG.

### 3.2. Batch Experiments with Imposed Power

Figure 6 presents the heating phase of a typical batch experiment with fixed power. The heating curves of water and ground blackcurrant pomace are presented. The mass of water plus biomass is constant in all the experiments.

The same protocol was used for both BCP and BSG. A constant power of 2 kW was applied for 700 or 800 s on the mixture and compared to the curve of water. The experiments were repeated at least twice. We observed that, initially, the temperature evolution was identical between the BCP and water cases. At about 220 °C, the curves diverge; the blackcurrant pomace shows accelerated heating up to about 260 °C. The difference in final temperature is the result of the additional energy liberated during the reaction. The calculated liberated energy does not appear correlated to the actual final temperature as long as this final temperature is above 260 °C. This suggests that most of the heat is liberated at a precise temperature, as suggested with results on experiments with an imposed temperature ramp.

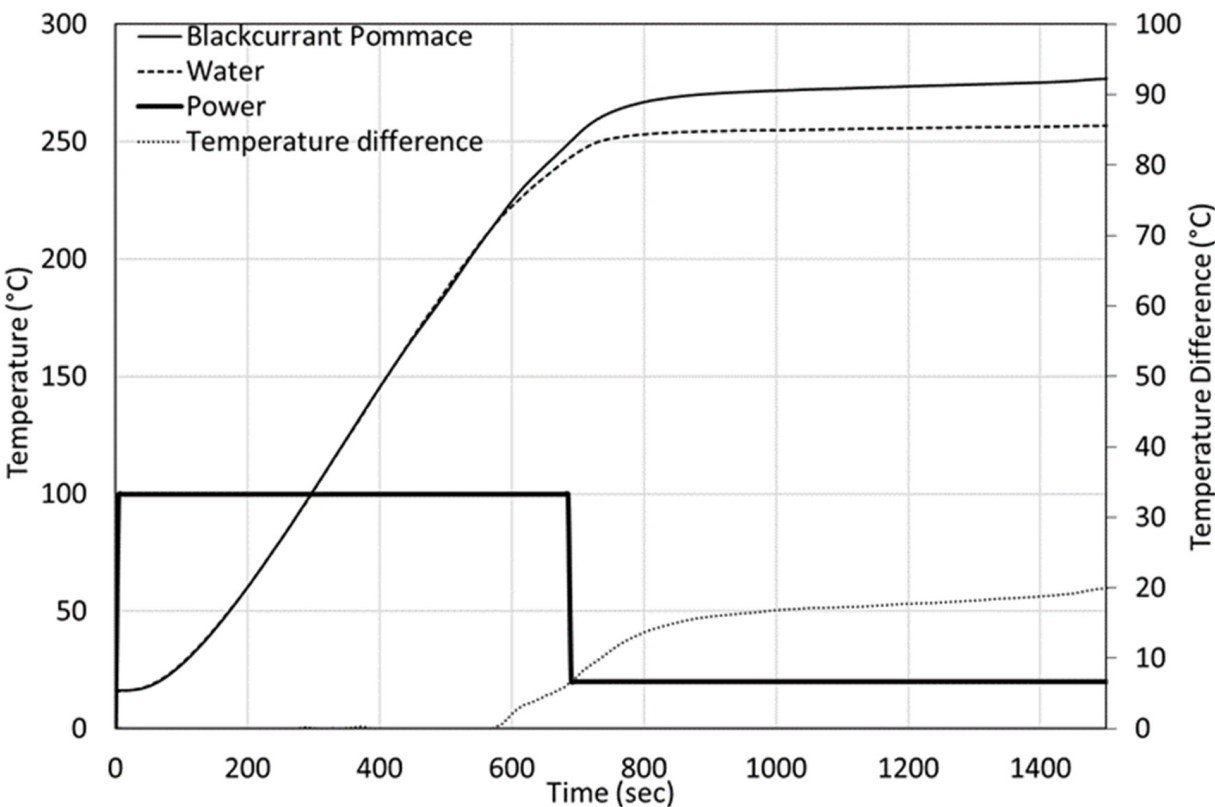

**Figure 6.** Applied power profile and temperature response of blackcurrant pomace compared to water.

The final temperature for the blackcurrant pomace case is significantly higher than the final temperature for water. The temperature difference is less pronounced with the brewers' spent grains case.

For BSG and BCP, there is a slight decrease in the heating rate of the reaction mixture between 200 and 270 °C, followed by acceleration at around 270 °C and onwards. While there is no clear explication of this behaviour, the same behaviour was observed for cellulose. Figure 7 presents the temperature evolution of cellulose and glucose in water, both at 15% concentrations. The curve for glucose presents the same behaviour that has been observed for blackcurrant pomace. The deviation between the water and the glucose occurs at a slightly higher temperature. Cellulose, however, reacts later and at 280 °C. The final temperature reached by the cellulose case is very close to the glucose case. We observed a similar behaviour that was seen for BSG, a sudden slowdown in the heating followed by acceleration. Cellulose takes longer to reach a final temperature similar to glucose. Longer residence time may have been beneficial for brewers' spent grains to observe stronger exothermic effects.

Other molecules, alkali lignin and glutamic acid, were tested with this method. Table 2 presents a summary of the obtained released energy for different model compounds. Care should be taken not to conclude too quickly from model molecules as actual biomass behaves quite differently than the sum of its constituents.

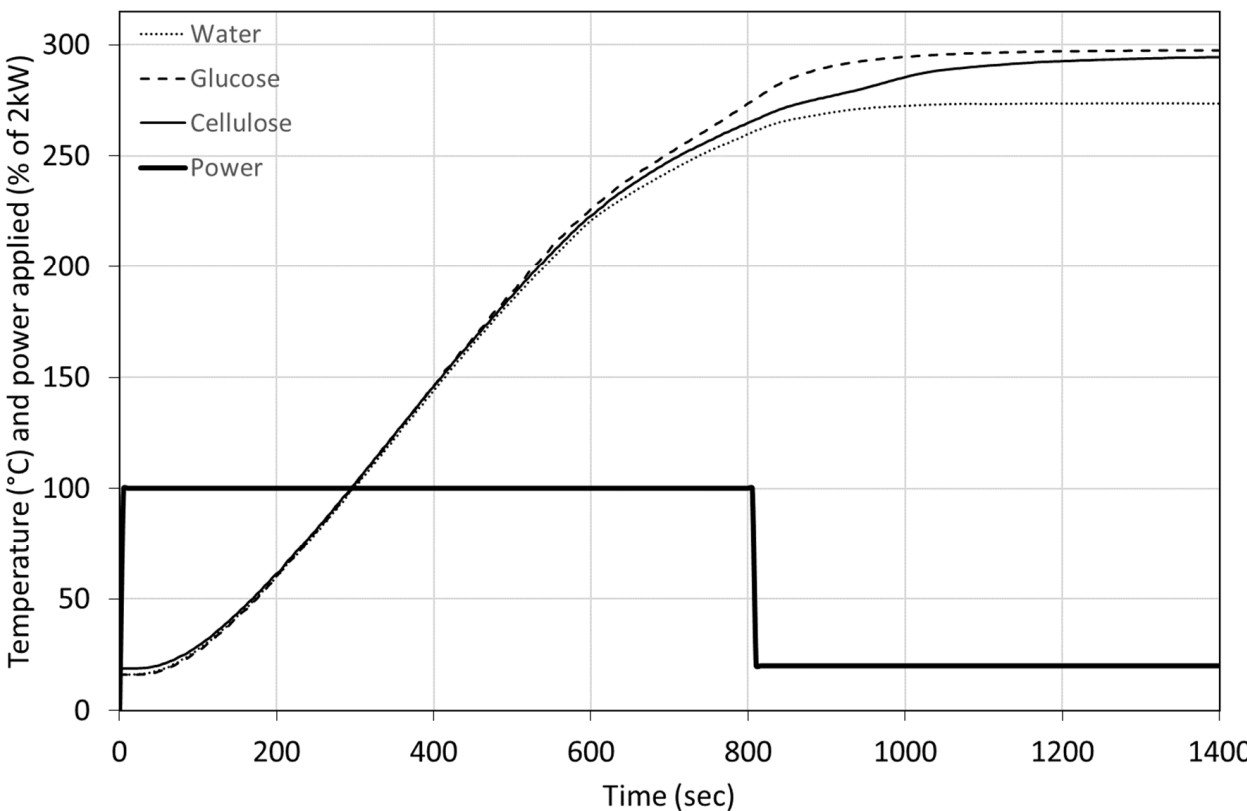

**Figure 7.** Applied power profile and temperature response of cellulose and glucose compared to water.

**Table 2.** Heat release in hydrothermal liquefaction conditions.

| Resource | Heat of Reaction (MJ/kg$_{biomass}$) | Standard Deviation (MJ/kg$_{biomass}$) |
|:---:|:---:|:---:|
| Blackcurrant pomace | −1.2 | 0.7 |
| Brewers' spent grains | −1.9 | 0.4 |
| Alkali lignin | 0.2 | 0.1 |
| Cellulose | −1.1 | 0.1 |
| Glucose | −1.3 | 0.05 |
| Glutamic acid | 0.4 | 0.2 |

We observe an exothermic behaviour for biomasses, cellulose and glucose. Even though BCP and BSG show a clear exothermic behaviour, the standard deviation remains high for the biomass cases.

### 3.3. Continuous Experiments

Figure 8 shows the evolution of the temperatures measured by the two first thermocouples. The first thermocouple (TC1) is near the inlet (opposite the reactor injection point), the second is 10 cm further downstream, separated by the first impeller (out of six). The temperature indicated by TC1 is generally lower than the other thermocouples as it is located near the inlet, and it is only used for information. TC1 reacts immediately after the injection of biomass and starts a ramp to a much higher value; it gains 34 °C in one hour and 14 min. It takes a while before descending to a stable value, 30 °C above the initial value. Stability is a relative notion for the continuous reactor as a true steady state is never reached. We can, however, infer a pseudo steady state from the averaged conditions.

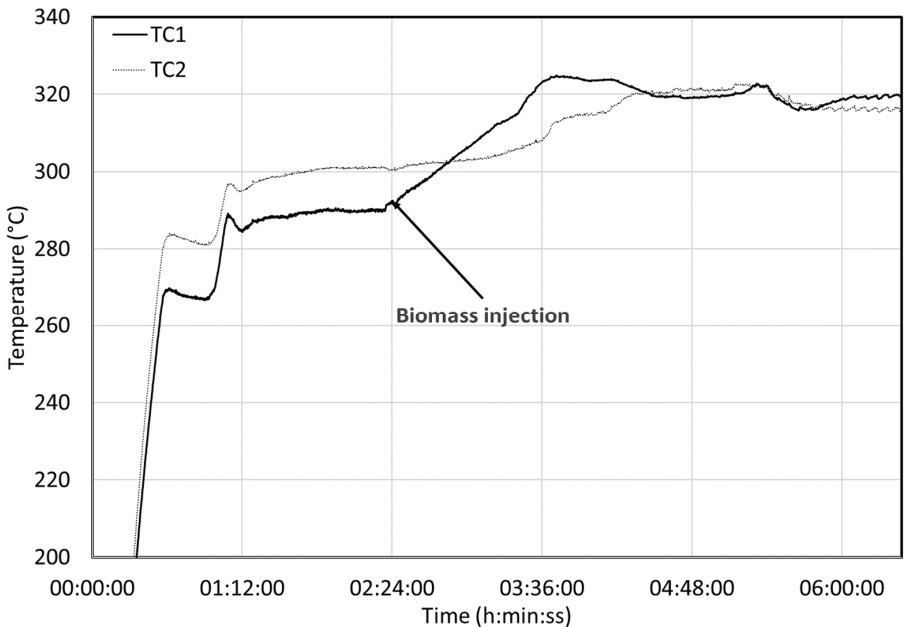

**Figure 8.** Temperature measurement on a continuous reactor with BCP.

The second thermocouple downstream (TC2) also registers the increase in temperature but much later and in an attenuated way compared to the TC1. Surprisingly, the final temperatures of TC1 and TC2 are very similar. The temperature increase of 31 °C represents a heat release of −1.98 MJ/kg$_{biomass}$. The attenuation of the temperature front shows that the hydrodynamics of the reactor are complex. Reactions occur very fast, so the first thermocouple is sensitive to heat release. It takes a little while to get to the second thermocouple due to the inertia of the reactor and the mixing.

We observed the same effect on BSG (Figure 9), as expected from batch experiments. Heat is released when the biomass is directly injected in a hot reactor. After a certain delay, the first thermocouple stabilized around 320 °C, as does the second thermocouple. The heat release can be estimated to be −1.97 MJ/kg$_{biomass}$.

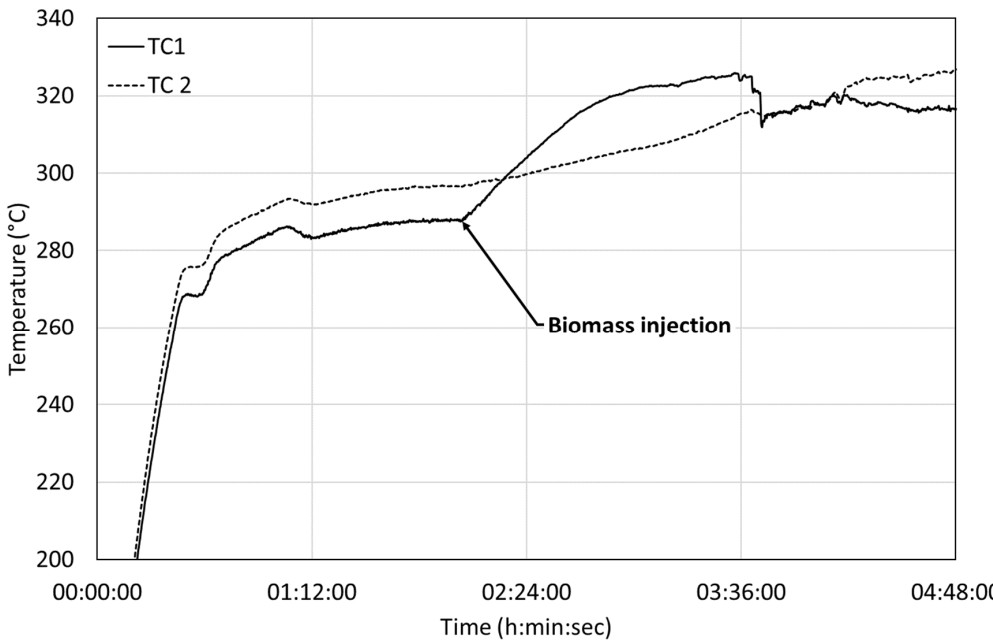

**Figure 9.** Temperature measurement on a continuous reactor with BSG.

### 3.4. Comparison of the Calculation of the Heat Release between Batch and Continuous Experiments

The reaction enthalpy has been calculated with Hess' law (see Supplementary Materials) from the enthalpy of formation of each product collected and their relative yields.

The enthalpy of formation of the gas is easily calculated as the sum of the enthalpy of formation of each gaseous compound observed in μ-GC and their molar fraction ($CO_2$ represents generally between 95 and 99% of the gas produced). The enthalpies of formation of biocrudes and the aqueous phases have been estimated though their higher heating value and the enthalpy of formation of the combustion products (see Supplementary Materials). The results of the calculation of the enthalpies of formation and the reaction enthalpies (by Hess' law) are given in Tables 3–5.

**Table 3.** Enthalpy of formation of BCP and products from the liquefaction of BCP.

| | $\Delta H_{f,molar}$ (MJ·mol$^{-1}$) | | $\Delta H_{f,mass}$ (MJ·kg$^{-1}$) | |
|---|---|---|---|---|
| | Continuous | Batch | Continuous | Batch |
| $\Delta H_{f,\,BCP}$ | 9.2 | | 7.4 | |
| $\Delta H_{f,biocrude}$ | 10.7 | $9 \pm 1$ | 9.1 | $8.4 \pm 0.8$ |
| $\Delta H_{f,aqueous\ phase}$ | 7.4 | 7.4 | 5.6 | 5.6 |
| $\Delta H_{f,gas}$ | $-0.4$ | $-0.4 \pm 0.1$ | $-8.8$ | $-8.8 \pm 0.1$ |

**Table 4.** Enthalpy of formation of BSG and products from the liquefaction of BSG.

| | $\Delta H_{f,molar}$ (MJ·mol$^{-1}$) | | $\Delta H_{f,mass}$ (MJ·kg$^{-1}$) | |
|---|---|---|---|---|
| | Continuous | Batch | Continuous | Batch |
| $\Delta H_{f,\,BSG}$ | 7.0 | | 5.4 | |
| $\Delta H_{f,biocrude}$ | 8.8 | $8.3 \pm 2$ | 7.5 | $7 \pm 2$ |
| $\Delta H_{f,aqeuous\ phase}$ | 7.0 | 5.8 | 5.3 | 4.4 |
| $\Delta H_{f,gas}$ | $-0.4$ | $-0.4 \pm 0.1$ | $-8.8$ | $-8.8 \pm 0.1$ |

**Table 5.** Reaction enthalpy for the liquefaction of BCP (Case at 300 °C, 15 min holding time) and BSG (Case at 315 °C, 15 min holding time).

| | BCP | | BSG | |
|---|---|---|---|---|
| | Continuous | Batch | Continuous | Batch |
| $\Delta H_{reaction}$ (MJ·kg$^{-1}$) | $-4.2$ | $-2.5 \pm 0.6$ | $-2.9$ | $-2 \pm 1$ |

The collection of products, especially biocrude, is essential to yield calculations. The viscous nature of the product is challenging and sometimes leads to losses in the pilot. As biomass is the unique source of carbon, the carbon balance is used to estimate the losses. In total, 86% of the carbon was recovered on the continuous run with the blackcurrant pomace and 95% was recovered on the continuous run with the brewers' spent grains. The high deviation between the values of batch and continuous experiments for BCP is due to the difficulties with collecting biocrude during continuous experiments. The biocrude tends to foul the heat exchanger, the transfer lines and the discharge pump.

The results are consistent with the exothermic behaviour observed previously. The values are high, as are the uncertainties due to the successive analysis and the propagation of the experimental errors.

This calculation does not take into account the enthalpy of formation of water. It is very difficult to dress an accurate mass balance for water. However, the enthalpy of formation of the organic fraction dissolved has been considered and influenced significantly the exothermic effect of the reaction. Without considering this fraction, the global heat release of the liquefaction reaction would have been overestimated to $-5.4$ MJ/kg$_{biomass}$ for blackcurrant pomace and $-4.2$ MJ/kg$_{biomass}$ for brewers' spent grains.

### 3.5. Synthesis on Reactivity

Exothermic reactions occur early in the heating curve (at around 200 °C) and continue over the span of up to 10 min. This suggests that exothermic effects may come from rapidly accessible reactants such as sugars (as for glucose) that can be released progressively with the hydrolysis of amorphous material such as hemicellulose in the case of BSG. The analysis of biocrudes highlights the dehydration taking place. Experiments on model compounds also highlight that decarboxylation can take place.

The amount of gas (mainly $CO_2$ > 90%) produced can be calculated from the pressure increase after reaction. Figure 10 presents the heat liberated as a function of the amount of $CO_2$ produced for each of the experiments.

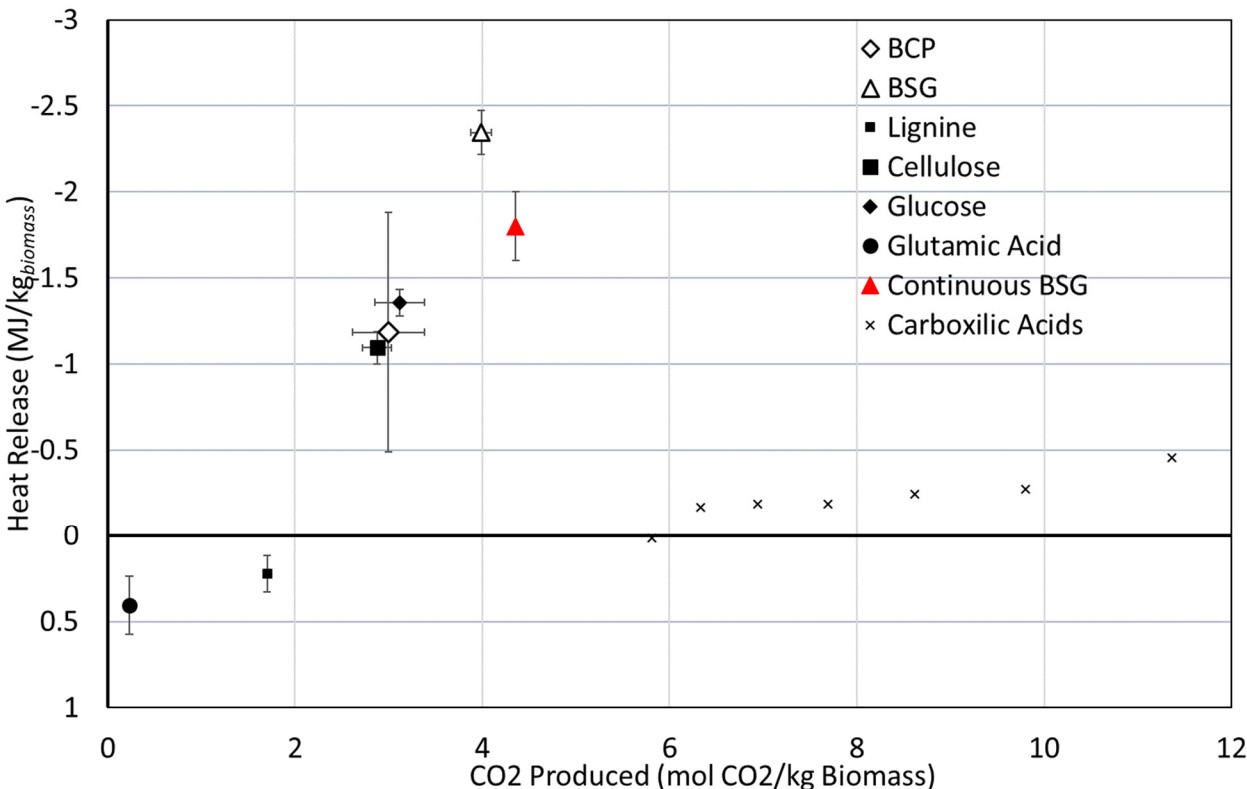

**Figure 10.** Relation between the heat release and the $CO_2$ produced.

There is no linear correlation between the amount of $CO_2$ and the amount of heat produced. This is to be expected as the resources are quite different. However, it seems that biomass containing sugars, cellulose or hemicellulose also produces more $CO_2$ and releases more heat.

Dehydration and decarboxylation reactions of some compounds were studied to check how much energy would be released from these reactions. Table 6 gives some examples.

**Table 6.** Heat of reaction (Calculated by Hess' law) of model compounds.

| Reaction | Heat of Reaction (MJ·kg$^{-1}$) |
|---|---|
| $C_3\text{-COOH} \rightarrow CO_2 + C_3\text{-H}$ | −0.46 |
| $C_6H_{12}O_6 \rightarrow C_6H_6O_3 + 3\ H_2O$ | −0.12 |
| $C_5H_{10}O_5 \rightarrow C_5H_4O_2 + 3\ H_2O$ | −0.02 |
| $C_5H_9NO_2 \rightarrow C_4H_7NO + H_2O + CO_2$ | 0.38 |

The values for the heat of formation are taken from the NIST Webbook [43]. It seems that exothermic behaviour cannot be fully explained by one reaction, it results from a sum of reactions, many of which are difficult to identify.

## 4. Discussion

All methods described here allow the estimation of the heat released during hydrothermal liquefaction. The estimation based on the characterisation of the products gives values with high uncertainties. A small deviation in analysis results in high uncertainties in the calculation of the released heat. In situ measurements limit the experimental error to one monitored parameter and are often more reliable.

Comparing the power curves for biomasses and water shows that there are two peaks in heat production. These peaks correspond to the peaks found by Ibbett et al. [33] for hemicellulose and cellulose at 228 and 260 °C, respectively.

By imposing a temperature ramp, we dispose of a very reproducible heating method regardless of the biomass. Comparison between batch runs is easy. The inconvenience of this method is that both temperature and power are free variables and not rigorously identical. Another problem is that the method only works up to the 100% output of the controller, just after most of the heat is released. Heat is probably released at a higher temperature. In this way, results in the heat release can be underestimated.

The second method is similar to the approach in a differential scanning calorimeter. Fixing the applied power profile is similar to an alternative method called differential thermal analysis (DTA). This method has the advantage in that it removes one uncertainty from the system, only the temperature needs to be measured. The minor inconvenience of the method is that the final temperature cannot be precisely predicted as this depends on the room temperature as well as the reactivity of the resources.

The results on model compounds show that the obtained values of the released heat for glucose and cellulose are close to those obtained earlier by Funke and Ziegler [24]. This was attributed to the decarboxylation and dehydration reactions occurring on cellulose and hemicellulose hydrolysis products. This also explains why Funke and Ziegler found hydrothermal carbonisation of wood to be less exothermic, due to its composition rich in lignin. Other reactions may contribute, such as polymerisation or condensation reactions from intermediates, and produce heavier compounds. Results have to be taken with caution though and further continuous experiments are needed.

It must be noted that model molecules, and in particular alkali lignin and microcrystalline cellulose, are not very representative of actual native lignin and cellulose. Crystallinity and interaction between native biomass compounds are assumed to play a role. Additionally, more reaction products are obtained with real biomass than with model molecules, suggesting that biomass is more complex and modelling of biomass is still a large field of investigation. Further characterisation and experiments on extractive biopolymers will be crucial in the comprehension of the transformation.

It also seems that kinetic limitations may play a role in dehydration reactions, probably due to the initial hydrolysis step. However, it appears that for temperatures above 250 °C, once sugars are released, the transformation occurs fast for the two biomasses. The fast character of these reactions and the temperature range under which heat is released support the assumption that decarboxylation and dehydration take part in the exothermic behaviour observed.

Blackcurrant pomace and brewers' spent grains have a water content of 50 and 75%, respectively. The heat that would be required for drying before potential combustion is 2.2 and 6.8 MJ/kg of dry matter, respectively, significantly higher than the heat released during the transformation. In practice, this is a moot point as these resources are not usually considered for energy applications. Further investigation on the separation of the biocrude with the aqueous phase is needed, even though this step is expected to be less energy consuming than biomass drying. The hydrothermal liquefaction process uses diluted biomass and displays a small exothermic behaviour under hydrothermal conditions,

up to −1 MJ/kg of dry biomass. This energy is liberated after heating the mixture from 20 to 300 °C at a cost of 12 MJ/kg dry matter. If the heat recovery is adequate, it may be possible to ensure full auto thermal operation.

## 5. Conclusions

This work presents a novel approach that completes many studies already published on hydrothermal liquefaction. The heat released for actual resources is generally between 1 and 3 MJ/kg dry matter in batch reactors. Lignin and glutamic acid show an exception with a slight endothermic behaviour. The results for blackcurrant pomace and brewers' spent grains show a slightly larger uncertainty than those for model compounds. The results in the continuous reactor are similar to those obtained in batch experiments.

Different methods of establishing the thermal effects of the liquefaction transformation have been explored in this paper. The method with constant power allows a more complete measure of exothermic behaviour. The resulting responses can be easily compared to evaluate the heat released from the reactions. This is the first time continuous experiments have exhibited thermal effects for liquefaction. These results confirm the observation of batch experiments.

Considering the biomass concentration of 10%, this would constitute 10% or even more of the heating requirement of the reaction mixture. The released heat will certainly help operations of a commercial plant to limit heat losses. The heat released will not be sufficient to allow an auto thermal operation, but the reaction energy could lower the energy demand in the case of an industrial process.

**Supplementary Materials:** The following are available online at https://www.mdpi.com/article/10.3390/chemengineering6010002/s1, Section S1.

**Author Contributions:** Conceptualization, M.B. and G.H.; methodology, M.B. and G.H.; resources, G.H.; data curation, G.H.; writing—original draft preparation, M.B., G.H. and A.R.; writing—review and editing, M.B., G.H., A.R. and P.F.; visualization, G.H.; supervision, P.F.; project administration, G.H.; funding acquisition, G.H. All authors have read and agreed to the published version of the manuscript.

**Funding:** This work was co-funded between CEA-LITEN and ADEME and realized using equipment of CEA-Grenoble. This research received no external funding.

**Data Availability Statement:** The data presented in this study are available on request from the corresponding author.

**Acknowledgments:** The help of Léa Vilcocq (CPE Lyon) with the chemical analysis is gratefully acknowledged.

**Conflicts of Interest:** The authors declare no conflict of interest.

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
