# Peer review of "Evaluation of the Heat Produced by the Hydrothermal Liquefaction of Wet Food Processing Residues and Model Compounds"

_2305-7084, doi:10.3390/chemengineering6010002_

Round 1
Reviewer 1 Report
The publication concerns the current topic. The work was written in the correct layout. The methods used are correct. The results are clearly presented.
Comments to be improved:
- Expand the literature review in the Introduction section.
I propose to view the publication: Knapczyk, A.; Francik, S.; Jewiarz, M.; ZawiÅ›lak, A.; Francik, R. Thermal Treatment of Biomass: A Bibliometric Analysis—The Torrefaction Case. Energies 2021, 14, 162. https://doi.org/10.3390/en14010162. This review contains many relevant publications that may be of assistance in this publication.
- In the chapter Material and Methods I propose to present the course of the experiment in the form of a flowchart, which will significantly increase its readability. Please complete the information on the number of repetitions, device model and calorimeter measuring range.
- ‘Error! Reference source not found..’- . ProszÄ™ o wyjaÅ›nienie czym jest to spowodowane.
- The chapter Discussion it looks like an extended chapter conclusions. There is no reference to the authors of other publications.
In the chapter Conclusion written: ‘Very few authors have published results on the thermal effects, or have attempted to quantify the heat released, of the hydrothermal liquefaction transformation. Most authors have limited their work to the basic conversion, yields, chemistry and the technical aspects of the process.’. Please provide examples of these publications.
Author Response
The publication concerns the current topic. The work was written in the correct layout. The methods used are correct. The results are clearly presented.
- Thank you, we have made significant improvements to the paper.
Point 1 - Expand the literature review in the Introduction section.
I propose to view the publication: Knapczyk, A.; Francik, S.; Jewiarz, M.; ZawiÅ›lak, A.; Francik, R. Thermal Treatment of Biomass: A Bibliometric Analysis—The Torrefaction Case. Energies 2021, 14, 162. https://doi.org/10.3390/en14010162. This review contains many relevant publications that may be of assistance in this publication.
- OK, we have added some more references, the introduction was thoroughly reviewed.
Point 2 - In the chapter Material and Methods I propose to present the course of the experiment in the form of a flowchart, which will significantly increase its readability. Please complete the information on the number of repetitions, device model and calorimeter measuring range.
- OK, a flow chart was added to better explain how the batch experimpents were analysed. We also included information on the number of repeats.
Point 3 - ‘Error! Reference source not found..’- . ProszÄ™ o wyjaÅ›nienie czym jest to spowodowane.
- ???
Point 4 - The chapter Discussion it looks like an extended chapter conclusions. There is no reference to the authors of other publications.
- For this paper we used the classic layout by having a results section, purely focussing on our work. In the separate discussion section the experimental result are placed in perspective by using some references. Many authors combine the results and discussion, up to the point in sometimes confusing the literature review with the experimental section, this is not good either.
Point 5 - In the chapter Conclusion written: ‘Very few authors have published results on the thermal effects, or have attempted to quantify the heat released, of the hydrothermal liquefaction transformation. Most authors have limited their work to the basic conversion, yields, chemistry and the technical aspects of the process.’. Please provide examples of these publications.
- Again, the literature survey is in the introduction. We are not a fan of referencing the literature in the conclusions.

Reviewer 2 Report
This is a interesting manuscript which focuses on the heat produced by the hydrothermal liquefaction of wet food processing residues. On the whole, this manuscript was well prepared by the authors. I suggest that the editor can accept this manuscript after minor revision.
Tables 3-5 were not cited in the text.
The section of conclusion should be simplified.
Author Response
This is a interesting manuscript which focuses on the heat produced by the hydrothermal liquefaction of wet food processing residues. On the whole, this manuscript was well prepared by the authors. I suggest that the editor can accept this manuscript after minor revision.
- Thanks, we have improved the paper with your comments and the comments of other reviewers.
Point 1 - Tables 3-5 were not cited in the text.
- Thanks, this is corrected
Point 2 - The section of conclusion should be simplified.
- Done

Reviewer 3 Report
This paper investigated the thermal effect during hydrothermal liquefaction of wet food residues and some model compounds. Two methods were used to evaluate the heat released in this process. Results showed that exothermic reactions happened during hydrothermal liquefaction of blackcurrant pomace, brewers’ spent grains and some model compounds such as cellulose and glucose, while the transformation of lignin and glutamic acid were endothermic. There is little work concerning the heat reaction in hydrothermal liquefaction, especially in continuous experiments, so this research is novel and will provide more insights into HTL and its industrial application. However, some flaws still exist and should be modified: (1) The title is insufficient to express the whole research content, studies of model compounds can also be present in the title; (2) Introduction is too verbose and should be refined, in addition, it seems more appropriate to reverse the sequence of the last two paragraphs. (3) Three methods of evaluating heat of reaction mentioned in introduction refers to batch experiments with temperature ramp, batch experiments with imposed power and continuous experiments with imposed power, is it more appropriate to classify them into two methods: constant heating ramp and constant power? (4) As an unclosed system, heat loss happens every minute during the experiment, how to evaluate the influence of heat diffusion between the reactor and circumstance to your results? (5) According to reference [31], H in equation (3) in 2.4 should be ln H, why do you use H instead? (6) Figure that describes the average curve on power response to temperature ramp controlled for batch experiments of brewers’ spent grains (BSG) and water is missing in 3.1. (7) There are too many gramma errors in this paper, modified traces are obvious in figure 2 and it should be present clearly in uniform sharpness, the format and number of titles inside the article is inconsistent and disordered, so comprehensive polish is suggested before this article is accepted.
Author Response
The title is insufficient to express the whole research content, studies of model compounds can also be present in the title;
- OK we changed the title to “Evaluation of the heat produced by the hydrothermal liquefaction of wet food processing residues and model compounds”
Introduction is too verbose and should be refined, in addition, it seems more appropriate to reverse the sequence of the last two paragraphs.
- OK, we made some modifications to the introduction to make it easier to read.
Three methods of evaluating heat of reaction mentioned in introduction refers to batch experiments with temperature ramp, batch experiments with imposed power and continuous experiments with imposed power, is it more appropriate to classify them into two methods: constant heating ramp and constant power?
- The methods are quite different and we prefer to maintain three different paragraphs in one section. We believe this makes it easier to understand.
As an unclosed system, heat loss happens every minute during the experiment, how to evaluate the influence of heat diffusion between the reactor and circumstance to your results?
- This is of course the principle to comparing the results to a reference case. We put the reactor in similar conditions with an inert mixture and compare it with a reactive mixture. Heat exchange effect should be mostly cancelled out. This is mentioned in paragraph 2.5 (new numbering)
According to reference [31], H in equation (3) in 2.4 should be ln H, why do you use H instead?
- Well spotted. Yes there was an error in the equation. Actually, the term should have included the log. It has no impact on the calculations.
Figure that describes the average curve on power response to temperature ramp controlled for batch experiments of brewers’ spent grains (BSG) and water is missing in 3.1.
- OK, the graph was replaced with a new one containing both BSG and BCP.
There are too many gramma errors in this paper, modified traces are obvious in figure 2 and it should be present clearly in uniform sharpness, the format and number of titles inside the article is inconsistent and disordered, so comprehensive polish is suggested before this article is accepted.
- The paper was thoroughly checked for errors. Figure 2 was affectively an edited version of an old graph, it was redrawn. The number of titles has been reduced.

Round 2
Reviewer 1 Report
The publication has been corrected and completed. The answers received are sufficient. I recommend the publication in its current form for publication in the ChemEngineering journal.